# Effects of Tool Edge Geometry on Chip Segmentation and Exit Burr: A Finite Element Approach

## Muhammad Asad 

Mechanical Engineering Department, Prince Mohammad Bin Fahd University, AL-Khobar 31952, Saudi Arabia; masad@pmu.edu.sa

**Abstract:** The effects of different tool edge geometries (hone and chamfer (T-land)) on quantitative measurement of end (exit) burr and chip segmentation (frequency and degree) in machining of AA2024-T351 are presented in this work. The finite element (FE) approach is adopted to perform cutting simulations for various combinations of cutting speed, feed, and tool edge geometries. Results show an increasing trend in degree of chip segmentation and end burr as hone edge tool radius or chamfer tool geometry macro parameters concerning chamfer length and chamfer angle increase. Conversely, the least effects for chip segmentation frequency have been figured out. Statistical optimization techniques, such as response surface methodology, Taguchi's design of experiment, and analysis of variance (ANOVA), are applied to present predictive models, figure out optimum cutting parameters, and their significance and relative contributions to results of end burr and chip segmentation. Various numerical findings are successfully compared with experimental data. The ultimate goal is to help optimize tool edge design and select optimum cutting parameters for improved productivity.

**Keywords:** tool edge preparation; segmented chip; machining simulation; burr; optimization

---

## 1. Introduction

Aluminum alloys are widely used in the aerospace industry due to their excellent strength-to-weight ratios and thermal properties. Aluminum alloys are categorized as easy to machine materials and are ideal candidates to subject to dry high-speed machining. However, certain complex combinations of tool materials, tool cutting angles (mainly rake angles), tool edge geometry (hone edge and chamfer edge), chip breaker profiles, cutting process parameters, machine dynamics, among others, greatly influence high-speed cutting processes and may result in high cutting temperatures and intense localized deformations, as reported in numerous experimental and numerical studies performed on aluminum alloys, such as AA2024-T351, AA7010-T7451, and AA7050-T7451. The severe cutting conditions lead to highly segmented chip morphology (higher "chip segmentation frequency" and higher "degree of chip segmentation"), poor surface finish, compromised surface integrity, along with high residual stresses and early failure of tools [1–5].

Furthermore, burr formation is another unlikely phenomenon associated with machining processes. Burr (the undesired and detrimental sharp material formed on workpiece edges) is formed during machining of metallic materials and composite/metal stacks in all sorts of machining processes, such as drilling, milling, turning, and broaching. However, ductile of machining materials generally results in pronounced burr lengths [6,7]. Deburring or burr removal is a necessary process before the component is ready for its functional life, providing the required surface quality and allowing integration into product assembly. Various mechanical, thermal, electrical, or chemical deburring processes employed in industry are costly, require technical expertise, and are quite time consuming [6,7]. These non-value-added post-machining deburring processes undermine the benefits of high-speed

machining of aluminum alloys. All of this necessitates the optimization of cutting parameters, tool materials, and angles and edge geometries to improve machined component quality, improve tool life, and eventually increase productivity. Worthy analytical, experimental, and numerical efforts have been carried out in this context to comprehend the chip formation process [8–12] and optimize cutting parameters to control surface quality and residual stresses [13–15]. Most recently, an integrated finite element and finite volume numerical model was presented by Hegab et al. [16] to analyze nano-additive-based minimum quantity lubrication (MQL) effects on machining forces, temperatures, and residual stresses. A considerable decrease in cutting temperatures and residual stress was reported using nano-additive-based MQL. This ultimately will help to increase tool life and improve surface integrity. Furthermore, physical comprehension of burr formation mechanisms and burr control through parametric optimization and tool and workpiece geometry optimization have also been widely discussed in literature [3,4,6,7,17–19].

The present work aims to examine the effect of tool edge geometry design (hone (round) edge and chamfer (T-land) edge), also called "tool edge preparation", on chip formation, chip segmentation frequency, degree of chip segmentation, and exit burr formation processes. Various combinations of two macro-level parameters of chamfer edge geometry, namely chamfer length ($l_\beta$) and chamfer angle ($\gamma_\beta$), and the macro geometry of the hone edge radius ($r_\beta$) are investigated (Figure 1, Section 2.1). Micro-level cutting edge geometry segments such as "cutting edge segment on flank face" and "cutting edge segment on rake face", as discussed by Denkena et al. [19], are not considered, as feed values taken in the current study are higher than the equivalent edge radii (Table 1, Section 2.1). Additionally, the workpiece material in the vicinity of the stagnation point (around which micro cutting geometry is defined by Denkena et al. [19]) is extremely deformed during machining and is removed during simulation after attaining the defined damage criteria (described later in Section 2.2).

To simulate chip segmentation and exit burr formation processes for orthogonal cutting of AA2024-T351 finite element analyses using various combinations of tool edge geometry, cutting speed and feed tests were performed. Higher values for the tool edge chamfer length ($l_\beta$), chamfer angle ($\gamma_\beta$), and hone edge radius ($r_\beta$) will certainly increase the negative rake angle in the vicinity of the stagnation point, and the increased workpiece area will experience high thermo-mechanical load. This will largely influence the primary shear zone, negative shear zone (responsible for exit burr formation), and material degradation, in turn reducing the augmentation of chip segmentation and leading to longer burr lengths. Chip segmentation and exit burr formation processes are the main focus of the present work due to their direct and indirect effects on machined surface quality and tool life. For example, chip segmentation frequency and degree of chip segmentation directly dictate residual stress patterns, intensity, and depth on machined surfaces [11,20]. The chip segmentation phenomenon also causes fluctuating cutting forces and harmful chatter vibration affecting machined surface and tool life [21–23], whereas burr not only influences machined surface quality but also influences the fatigue life of machined parts [4,6,7]. A phenomenal shift from "thermal softening" to "crack initiation and propagation" has also been highlighted [12,21,24], causing formation of segmented chips using varying tool edge geometries, cutting speeds, and feeds. This paper also provides more comprehensive information on burr formation ("negative burrs" at the exit end of workpiece), crack propagation at the front of the tool edge, formation of negative shear zones and pivot point locations, boot-type chip formation, and associated burr generation phenomena. The eventual aim of the presented work is to provide further insight into chip and burr formation in machining of AA2024-T351 and to optimize cutting parameters and tool edge design for improved productivity, employing a finite element (FE)-based design and analysis approach. Numerically computed results of chip morphology, cutting forces, and chip segmentation frequency are compared with the ones obtained previously by performing orthogonal cutting experimental investigations on AA2024-T351 under similar cutting conditions [11].

A full factorial Taguchi's design of experiment (DOE) technique is employed to determine optimum combinations of tool edge geometry, cutting speed, and cutting feed to curtail burr lengths, chip

segmentation frequency, and degree of chip segmentation. Analysis of variance (ANOVA) is performed to determine the percentage influence of these factors on exit burr lengths, segmentation frequency, and degree of segmentation. Response surface methodology (RSM)-based quadratic predictive models are also proposed.

## 2. Finite Element Based Orthogonal Cutting Model

### 2.1. Geometrical Model, Mesh, Constraints, and Hypothesis

Figure 1 shows workpiece and tool geometrical models for orthogonal cutting cases, conceived in Abaqus explicit software (Abaqus, 6.16, Dassault Systemes, Johnston, RI, USA, 2016). For the present work, six different cutting edge geometries are considered: two hone edge ($r_\beta$ = 5 μm and 20 μm) and four chamfer edge (chamfer length ($l_\beta$) = 0.1 mm, chamfer angle ($\gamma_\beta$) = 15°; $l_\beta$ = 0.1 mm, $\gamma_\beta$ = 25°; $l_\beta$ = 0.2 mm, $\gamma_\beta$ = 15°; $l_\beta$ = 0.2 mm, $\gamma_\beta$ = 25°) technologies. In the current work, chip separation is based on ductile damage of a predefined sacrificial material layer approach [11], named the "chip separation zone" in Figure 1. The width of the "chip separation zone" is kept to the order of the tool hone edge radius ($r_\beta$), as per experimental evidence [25]. For hone edge radii of 5 μm and 20 μm, the "chip separation zone" width is taken as 20 μm, while for chamfer edge geometries, the "chip separation zone" is taken as the "equivalent radius ($r_{eq}$)" of chamfer edge geometries, as shown in Figure 1 and summarized in Table 1.

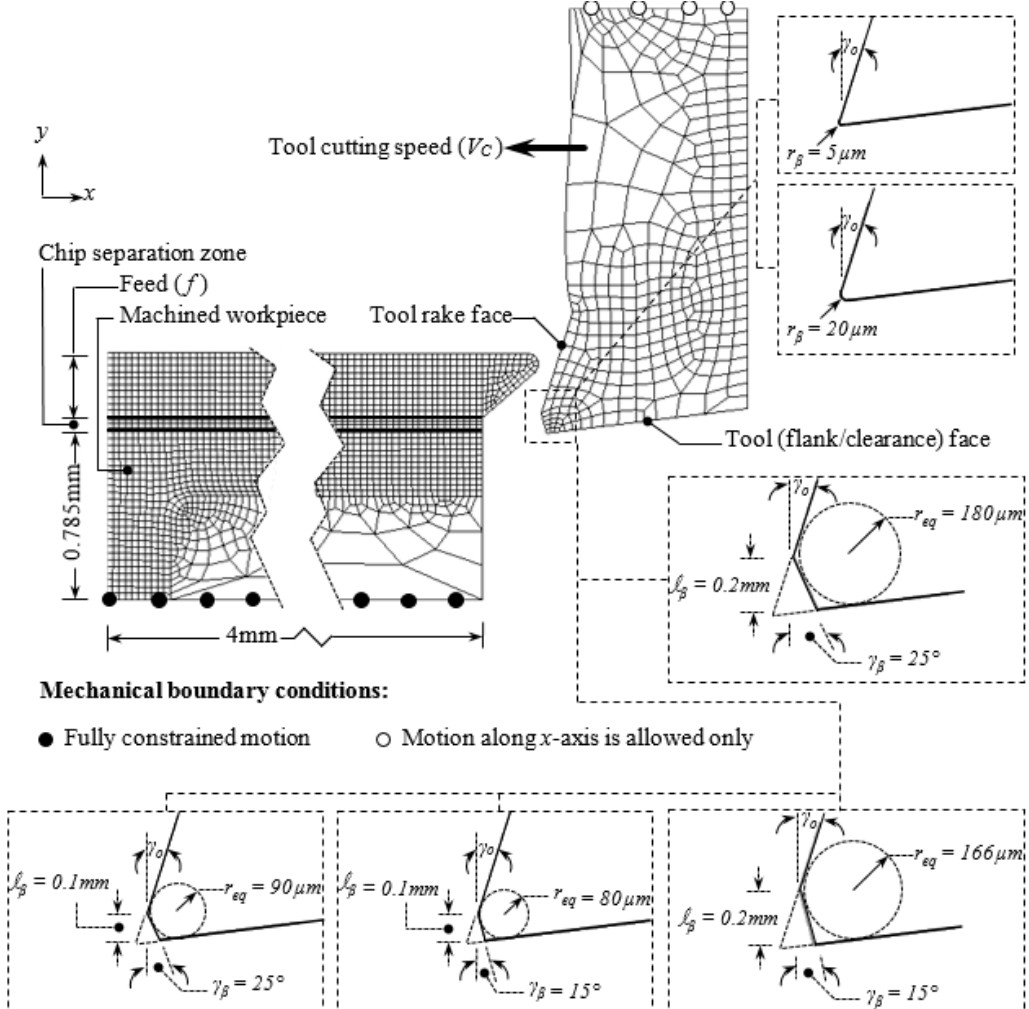

**Figure 1.** Orthogonal geometrical model and constraints.

**Table 1.** The "chip separation zone" width for various tool edge geometries.

| Tool Edge Geometry | Equivalent Radius, $r_{eq}$ (µm) | "Chip Separation Zone" Width (µm) |
|---|---|---|
| Hone edge ($r_\beta$ = 5 µm) | 5 | 20 |
| Hone edge ($r_\beta$ = 20 µm) | 20 | 20 |
| Chamfer length ($l_\beta$) = 0.1 mm, chamfer angle ($\gamma_\beta$) = 15° | 80 | 80 |
| Chamfer length ($l_\beta$) = 0.1 mm, chamfer angle ($\gamma_\beta$) = 25° | 90 | 80 |
| Chamfer length ($l_\beta$) = 0.2 mm, chamfer angle ($\gamma_\beta$) = 15° | 166 | 170 |
| Chamfer length ($l_\beta$) = 0.2 mm, chamfer angle ($\gamma_\beta$) = 25° | 180 | 170 |

In the FE model, the tool rake angle = 17.5° and the clearance angle = 7°, and the profile of insert chip breaker geometry are obtained using scanning electron microscope (SEM: Zeiss SUPRA 55-VP FEGSEM, Oberkochen, Germany) and are similar to that of Sandvik's "uncoated carbide insert: CCGX 12 04 08-AL 93 H10 (Sandvik Coromant Sandviken, Sweden)" geometry used in experimental work [11]. The workpiece geometry is modeled initially in three parts: the "machined workpiece", "chip separation zone", and the chip (with specific feed, *f*). Later on, parts are assembled, as per Figure 1, with the Abaqus built-in tie constraint algorithm, which ensures that all parts behave as a single entity during simulation. The objective for generating distinct parts (the "machined workpiece", "chip separation zone", and chip) lies in the ease of defining different material behaviors and governing equations in different sections of the workpiece.

During machining, heat is generated due to plastic work and friction at the tool and workpiece interface; therefore, to perform coupled temperature–displacement simulations, both the tool and workpiece are meshed with four-node, bilinear, quadrilateral continuum, displacement and temperature, reduced integration elements (CPE4RT), using the plane strain hypothesis. In these elements, along with displacement, temperature is also a nodal variable. Selection of an optimum mesh density in metal machining simulation producing physical results is quite challenging because of the non-availability of a specifically defined criterion in the literature. However, as a general rule, the finer the mesh, the higher the cutting force due to the size effect phenomenon [2]. A mesh sensitivity analysis for various mesh densities (Figure 2) was performed for *f* = 0.4 mm/rev and $V_C$ = 100 m/min. The increase in cutting forces as a function of mesh density can be figured out. An asymptotic value of mesh size of approximately 25 µm was achieved. Any further decrease in mesh density will not change cutting forces considerably, however, it will attract a time penalty in numerical simulation. A mesh density in the order of 20 µm is chosen in the "chip separation zone", chip, and upper layer (~0.3 mm) of the machined workpiece. The workpiece is fully constrained, while the tool advances with defined cutting speed in the x-direction during simulation, as shown in Figure 1. Cutting simulations were performed with twenty-four various combinations of cutting speed ($V_C$), feed (*f*), and tool edge geometries (Table 2).

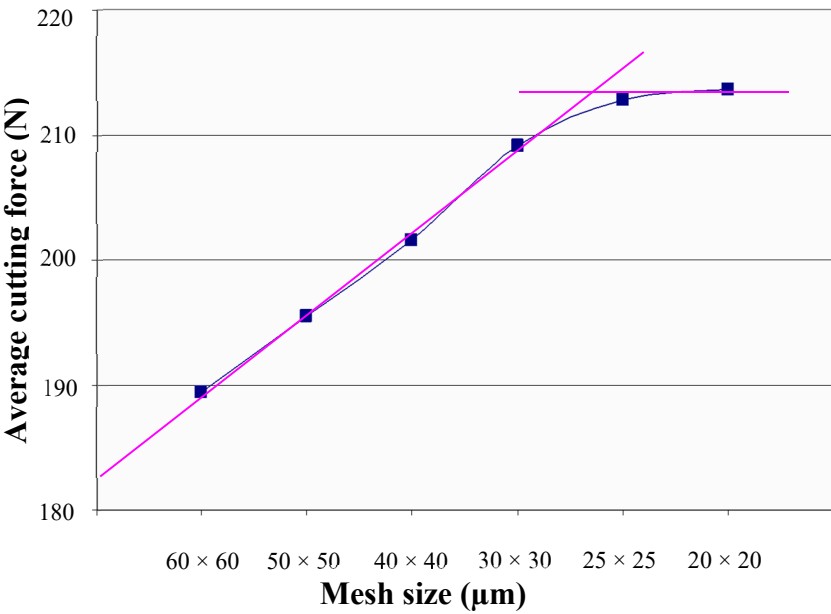

**Figure 2.** Average cutting force (N) for various mesh densities (μm) for plain strain conditions.

**Table 2.** Levels of cutting parameters.

| Level | Factors | | |
|---|---|---|---|
| | Tool Edge Equivalent Radius, $r_{eq}$ (μm) | Cutting Speed, $V_C$ (m/min) | Feed Rate, $f$ (mm/rev) |
| 1 | 5 | 800 | 0.3 |
| 2 | 5 | 400 | 0.3 |
| 3 | 20 | 800 | 0.3 |
| 4 | 20 | 400 | 0.3 |
| 5 | 80 | 800 | 0.3 |
| 6 | 80 | 400 | 0.3 |
| 7 | 90 | 800 | 0.3 |
| 8 | 90 | 400 | 0.3 |
| 9 | 166 | 800 | 0.3 |
| 10 | 166 | 400 | 0.3 |
| 11 | 180 | 800 | 0.3 |
| 12 | 180 | 400 | 0.3 |
| 13 | 5 | 800 | 0.4 |
| 14 | 5 | 400 | 0.4 |
| 15 | 20 | 800 | 0.4 |
| 16 | 20 | 400 | 0.4 |
| 17 | 80 | 800 | 0.4 |
| 18 | 80 | 400 | 0.4 |
| 19 | 90 | 800 | 0.4 |
| 20 | 90 | 400 | 0.4 |
| 21 | 166 | 800 | 0.4 |
| 22 | 166 | 400 | 0.4 |
| 23 | 180 | 800 | 0.4 |
| 24 | 180 | 400 | 0.4 |

*2.2. Material Behavior, Chip Separation, Friction, and Thermal Models*

The workpiece material's behavior is defined by the Johnson–Cook thermo-elasto-visco-plastic constitutive model (Equation (1)). This law adequately defines material behavior in high-speed metal deformation applications. Chip formation and separation are based on the evolution of ductile fracture [5]. The Johnson–Cook shear damage model (Equation (2)) is used to simulate ductile damage.

Initially, Equation (3) is used to calculate scalar damage initiation. Then, modeling of damage evolution is based on Equation (4), representing the linear evolution of scalar damage evolution parameter ($D$), and Equation (5), representing the exponential evolution of scalar damage evolution parameter ($D$). Equations (4) and (5) are used in chip separation and chip regions, respectively. In the latter equation, $G_f$, represents the fracture energy required to open the unit area of a crack, as per Hillerborg et al.'s fracture energy proposal [26], and is considered a material property. As per the approach, the material softening response after damage initiation is characterized by a stress–displacement response rather than a stress–strain response, and fracture energy is then given as Equation (6). In the present work, $G_f$ is taken as an input material parameter calculated by Equation (7). Finally, Equation (8) is used to calculate the equivalent plastic displacement at failure.

$$\overline{\sigma}_{JC} = \underbrace{(A + B\overline{\varepsilon}^n)}_{Elasto-plastic\ term} \underbrace{\left[1 + Cln\left(\frac{\dot{\overline{\varepsilon}}}{\dot{\overline{\varepsilon}}_0}\right)\right]}_{Viscosity\ term} \underbrace{\left[1 - \left(\frac{T - T_r}{T_m - T_r}\right)^m\right]}_{Softening\ term} \tag{1}$$

$$\overline{\varepsilon}_{0i} = \left[D_1 + D_2 exp\left(D_3\frac{P}{\overline{\sigma}}\right)\right]\left[1 + D_4 ln\left(\frac{\dot{\overline{\varepsilon}}}{\dot{\overline{\varepsilon}}_0}\right)\right]\left[1 + D_5\left(\frac{T - T_r}{T_m - T_r}\right)\right] \tag{2}$$

$$\omega = \sum \frac{\Delta\overline{\varepsilon}}{\overline{\varepsilon}_{0i}} \tag{3}$$

$$D = \frac{\overline{u}}{\overline{u}_f} \tag{4}$$

$$D = 1 - exp\left(-\int_0^{\overline{u}} \frac{\overline{\sigma}}{G_f} d\overline{u}\right) \tag{5}$$

$$G_f = \int_0^{\overline{u}_f} \sigma_y d\overline{u} \tag{6}$$

$$\left(G_f\right)_{I,II} = \frac{1 - v^2}{E}\left(K_C^2\right)_{I,II} \tag{7}$$

$$\overline{u}_f = \frac{2G_f}{\sigma_y} \tag{8}$$

During the progression of material damage, as the damage evolution parameter ($D$) approaches a value of one, it is assumed that the element's stiffness is fully degraded and that it can be removed from the mesh. Hence, chip separation from the workpiece body is realized. The tool (tungsten carbide) is modeled as a purely elastic body in the present work. Tool and workpiece material properties and model equation parameters are shown in Tables 3 and 4, respectively.

**Table 3.** Physical properties of tool and workpiece materials [11].

| Parameters | Workpiece (AA2024-T351) | Insert (Tungsten Carbide) |
|---|---|---|
| Density, $\rho$ | 2700 | 11,900 |
| Young's modulus, $E$ | 73,000 | 534,000 |
| Poisson's ratio, $v$ | 0.33 | 0.22 |
| Fracture energy, $G_f$ | $20 \times 10^3$ | X |
| Specific heat, $C_p$ | $0.557\,T + 877.6$ | 400 |
| Expansion coefficient, $\alpha_d$ | $8.91^{-3}\,T + 22.2$ | X |
| Thermal conductivity, $\lambda$ | $25 \leq T \leq 300\text{: } \lambda = 0.247T + 114.4$ $300 \leq T \leq T_m\text{: } \lambda = -0.125T + 226$ | 50 |
| Meting temperature, $T_m$ | 520 | X |
| Room temperature, $T_r$ | 25 | 25 |
| Fracture toughness ($K_{IC}$ and $K_{IIC}$) | 26 and 37 | X |

**Table 4.** Johnson–Cook model parameters for AA2024-T351 [11].

| $A$ | $B$ | $n$ | $C$ | $m$ | $D_1$ | $D_2$ | $D_3$ | $D_4$ | $D_5$ |
|---|---|---|---|---|---|---|---|---|---|
| 352 | 440 | 0.42 | 0.0083 | 1 | 0.13 | 0.13 | −1.5 | 0.011 | 0 |

During the machining process, heat is produced due to friction and plastic work. Conduction is the only mode of heat transfer considered in the present work, while the definition of contact conductance between the tool and workpiece ensures thermal conduction between them. Heat generation due to plastic work is modeled via Equation (9).

$$\dot{q}_p = \eta_p \overline{\sigma}.\dot{\overline{\varepsilon}} \tag{9}$$

where $\dot{q}_p$ is the heat generation rate due to plastic deformation and $\eta_p$ is the plastic (inelastic) heat fraction, taken as equal to 0.9. The heat generation rate due to friction is calculated by employing Equation (10).

$$\dot{q}_f = \rho C_P \frac{\Delta T_f}{\Delta t} = \eta_f\, J\, \tau_f\, \dot{\gamma} \tag{10}$$

An amount of heat $J$ (from the fraction of dissipated energy $\eta_f$ caused by friction) remains in the chip $(1 - J)$ and is conducted to the tool. The fraction of heat $J$ is a function of conductivities and diffusivities of tool and workpiece materials [27]. These thermal properties are temperature-dependent (Table 3) and vary with tool and workpiece contact during highly dynamic cutting processes. All of this makes it quite challenging to consider an accurate value of $J$ for tool–workpiece contact. Therefore, in the present work the Abaqus default value of $J = 0.5$ is taken. The steady state, two-dimensional form of the energy equation is given by Equation (11).

$$\lambda \left( \frac{\partial^2 T}{\partial x^2} + \frac{\partial^2 T}{\partial y^2} \right) - \rho C_p \left( u_x \frac{\partial T}{\partial x} + u_y \frac{\partial T}{\partial y} \right) \dot{q}_f + \dot{q}_p = 0 \tag{11}$$

Accurate and precise definition of friction characteristics between the tool and workpiece is important as well as challenging, since it depends on tool and workpiece material properties and geometries, cutting temperature, cutting speed, contact pressure, cutting forces, and contact length, among others [28,29]. Valuable research studies have been dedicated to this important aspect of metal machining to develop a more precise and realistic friction model under variable cutting conditions, owing to its importance in affecting the chip geometry, built-up edge formation, cutting temperature, tool wear, and surface integrity, among others. Application of these friction models in finite-element-based machining models can be taken into account when numerical models are based on the Eulerian formulation; nevertheless, it is still challenging when numerical models are based on the Lagrangian formulation. In the finite element cutting models based on the latter formulation, the workpiece mesh experiences high deformation in the vicinity of the tool–workpiece interaction. Simultaneously, when

damage and fracture energy approaches are used in constitutive models, the contact conditions become highly dynamic and complex. As the present work is based on the Lagrangian formulation, to avoid complexities in simulation, a basic Coulomb's fiction law has been adopted.

## 3. Results and Analysis

### 3.1. Finite Element Analysis and Discussion

Coupled temperature displacement cutting simulations for 24 combinations of feed, cutting speed, and tool edge geometries were performed, as per Table 2. Computational results concerning cutting forces, chip segmentation frequency, chip segmentation intensity, temperature distribution in the workpiece and tool, and end (exit) burr are calculated. Results of average cutting forces, chip morphology, and chip segmentation frequency (with tool edge equivalent radius, $r_{eq} = 20$ μm) are compared with the related available results of the experimental work [11]. Numerical results of cutting forces are found to have good correlation with the related experimental ones, as shown in Table 5. The results of chip segmentation frequencies for levels 15 and 16 ($V_C = 800$ m/min, $f = 0.4$ mm/rev, $r_{eq} = 20$ μm and $V_C = 400$ m/min, $f = 0.4$ mm/rev, $r_{eq} = 20$ μm) adequately correspond to their experimental counterparts. However, chip segmentation frequencies for levels 3 and 4 ($V_C = 800$ m/min, $f = 0.3$ mm/rev, $r_{eq} = 20$ μm and $V_C = 400$ mm/min, $f = 0.3$ mm/rev, $r_{eq} = 20$ μm) do not correspond well. The latter is due to the fact that at lower cutting feeds, segmentation intensity decreases (i.e., more uniform chip or less intense segmented chip morphology results). A more refined mesh would be required to obtain more accurate "segmentation frequency" results at lower cutting feeds, which would attract a greater time penalty in numerical simulations. Numerical findings (as presented in Table 5 and Figure 3a) only at levels 3, 4, 15, and 16 are compared with available experimental data results [11]. This comparison is made to validate the numerical model, whereas the rest of the numerical simulations made with various combinations of speed, feed, and tool edge geometry (levels 1, 2, 5–14, and 17–24) are merely exploitation of the validated numerical model (with no experimental results found in the literature). Numerically simulated and experimentally acquired chip morphologies (level 15 only) are compared in Figure 3.

**Table 5.** Numerical and experimental [11] comparison of mean cutting forces (at constant cutting depth, $a_P = 4$ mm) and chip segmentation frequencies.

| Levels | Numerically Computed Cutting Forces (N) | Experimentally Registered Cutting Forces (N) | Numerically Computed Chip Segmentation Frequency (kHz) | Experimentally Registered Chip Segmentation Frequency (kHz) |
|---|---|---|---|---|
| 3 | 669 | 769 | 53 | 90 |
| 4 | 657 | 769 | 26 | 37 |
| 15 | 840 | 976 | 63 | 65 |
| 16 | 833 | 978 | 31 | 32 |

### 3.1.1. Cutting Parameters and Tool Geometry Effects on Chip Segmentation Frequency and Segmentation Intensity

In almost all parametric combinations of cutting speed, feed, and tool edge radius, a slightly segmented to highly segmented chip morphology is reported. This shows the high plasticity properties of the alloy. Segmented chips (with high segmentation frequency and segmentation intensity) negatively affect machined surface integrity in terms of the quality of the surface profile, residual stress patterns, and the intensity of residual stresses. In the literature, these chips were also reported to produce periodic fluctuations in cutting forces and tool vibrations, which eventually effect tool life. The mechanism of formation of segmented chips is still not well understood, owing to the complex nature of the machining process, which is greatly influenced by the material properties and microstructure, tool geometries, cutting parameters, machine tool dynamics, and friction, among others [12,30]. However,

there are mainly two theories explaining the phenomenon of chip segmentation in most of the ductile materials: (a) thermoplastic deformation and formation of adiabatic shear bands because of thermal softening; and (b) fracture, where cracks initiate and propagate in the primary shear zone [12]. In the present work, both phenomena have been witnessed.

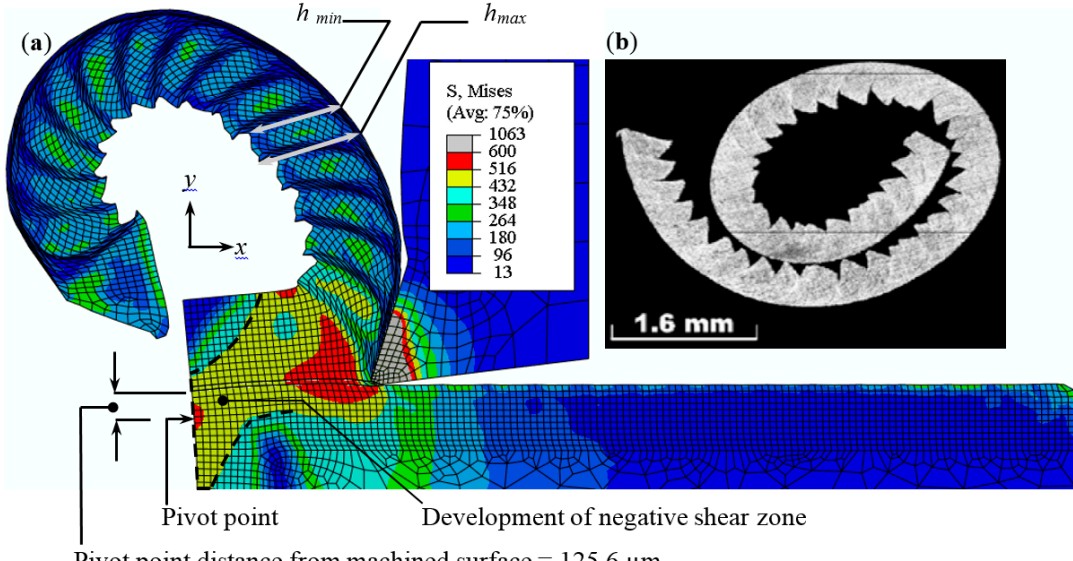

**Figure 3.** Chip morphology for cutting speed $V_C$ = 800 m/min, $f$ = 0.4 mm/rev, $r_{eq}$ = 20 μm (level 15): (**a**) numerically simulated; (**b**) experimentally generated.

At high cutting speed, frictional resistance causes an increase in cutting temperatures at the tool–workpiece interface, resulting in thermal softening (Figure 4). The thermal softening phenomenon dominates strain hardening, the material stiffness degrades (lower stresses in the vicinity of the tool edge; Figure 3a), and the material flows in the primary shear zones with ease, leading to generation of adiabatic shear bands. Apart from obvious results of higher cutting temperatures due to higher cutting speed, it can also be seen from Figure 4 that an increase in tool edge radius (especially tools with chamfer geometry) results in lower cutting temperatures. Similar trends have also been reported by Ozel [31] for cutting of AISI H-13 with cubic boron nitride (CBN) cutting inserts. This phenomenon is due to the size effect (i.e., more specific cutting energy is required as the tool radius increases in comparison to uncut chip thickness). A wider area now experiences plastic deformation, which requires more energy, and more heat is generated. However, the heat due to inelastic work is more easily dispersed over a large surface area with a larger equivalent edge radius, and consequently maximum temperatures are lower. At higher feed, higher temperatures are produced due to larger amount of plastic work (Figure 4). However, the rate of increase of temperature is not high enough (for feed variation studied in this work this ranges from 0.3 to 0.4 mm/rev) to cause any considerable thermal softening. Furthermore, at higher feed values, due to length effect, longer segments of chips are generated (i.e., frequency of segments will decrease). This shows that higher cutting speeds supplemented with a lower feed rate and lower tool edge radius promote formation of more adiabatic shear bands (high frequency of segmented chip morphology), mainly due to thermal softening. Segmentation frequency is greatly influenced by variation of cutting speed, while segmentation intensity or degree of chip segmentation, calculated by "$(h_{max} - h_{min})/h_{max}$", seems to be least effected by speed variation, as can be seen in Figure 5.

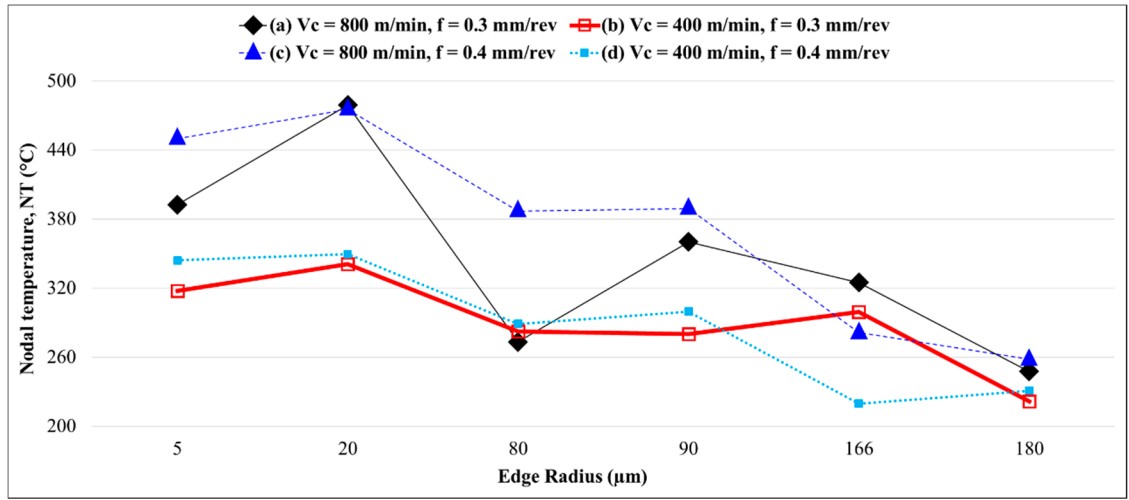

**Figure 4.** Maximum nodal temperature evolution for cutting speeds, feed, and tool edge radius variations.

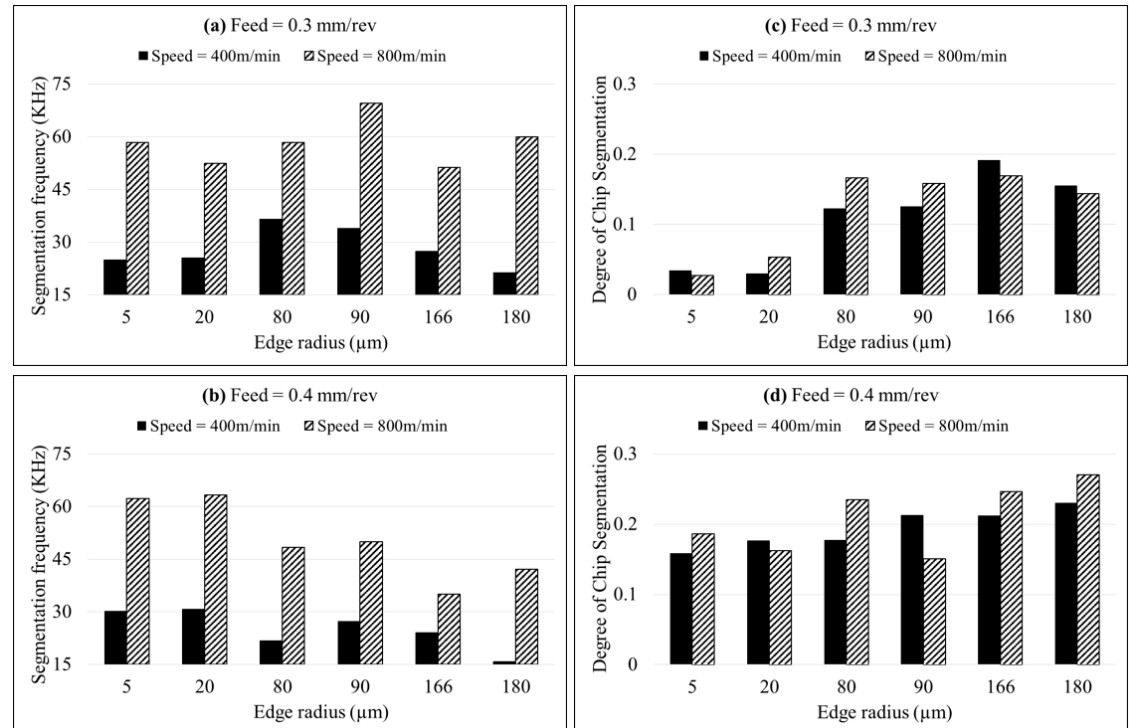

**Figure 5.** Cutting speed effect on segmentation frequency and degree of chip segmentation: (**a**) Segmentation frequency for $f$ = 0.3 mm/rev; (**b**) Segmentation frequency for $f$ = 0.4 mm/rev; (**c**) Degree of chip segmentation for $f$ = 0.3 mm/rev; (**d**) Degree of chip segmentation for $f$ = 0.4 mm/rev.

Figure 6 shows that an increase in cutting edge radius rarely influences the segmentation frequency, which largely influences the degree or intensity of segmentation. Indeed, as the chamfer tool angle ($\gamma_\beta$) increases, the effective rake angle in the vicinity of the stagnation point becomes more negative, and as the chamfer tool length ($l_\beta$) or hone edge radius ($r_\beta$) increase, the workpiece area experiences high thermo-mechanical load, leading to initiation and propagation of fracture in the primary shear zone. Furthermore, it can be noticed that chamfer tool length ($l_\beta$) contributes more than chamfer tool angle ($\gamma_\beta$) in intensifying the degree of segmentation and the equivalent edge radius (Table 1). On the other hand, as discussed previously and depicted in Figure 4, the increase in cutting edge radius results in

decreasing temperature; hence, thermal softening is not the dominant or responsible mechanism for chip segmentation at higher values of tool cutting edge radii.

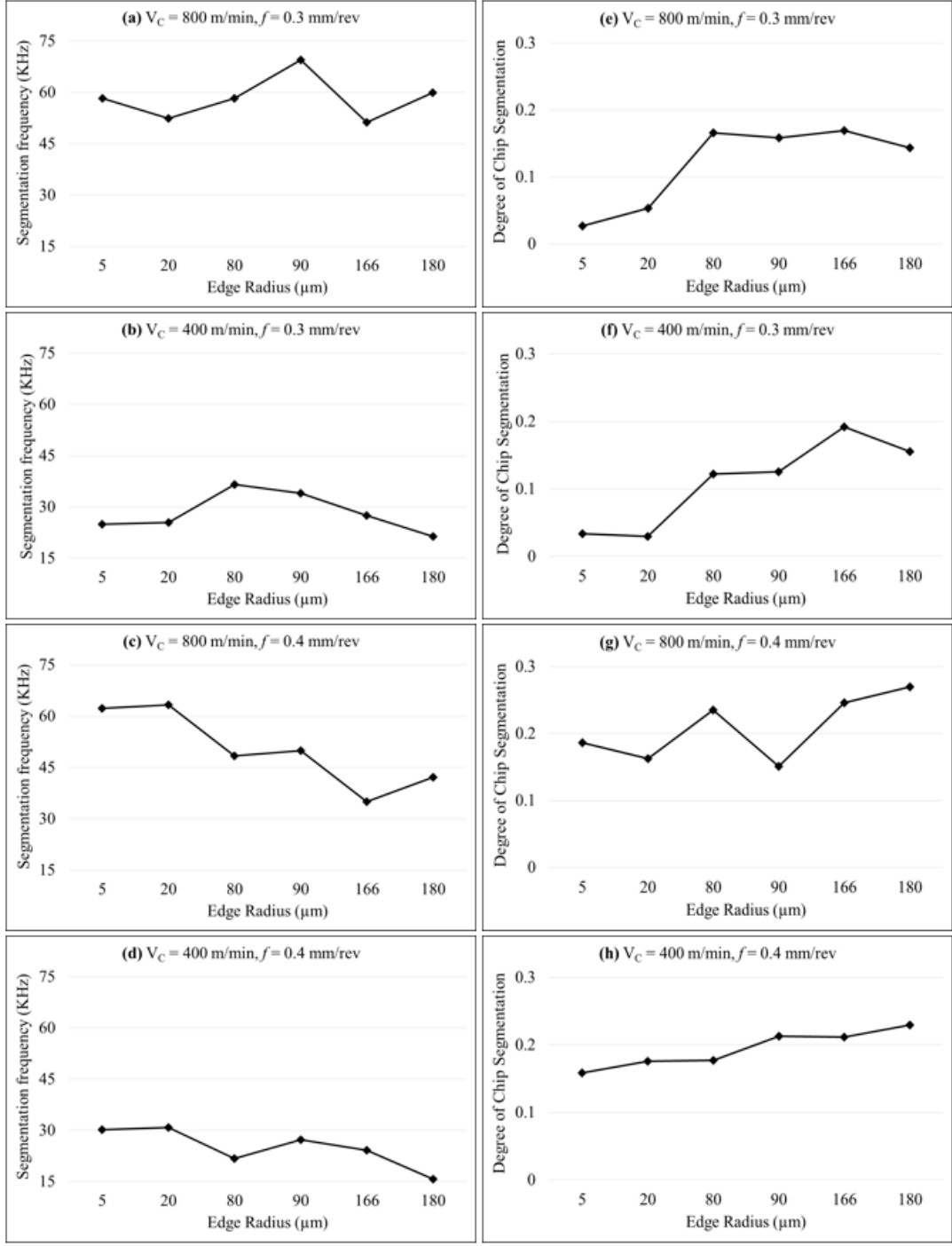

**Figure 6.** Edge radius effect on segmentation frequency and degree of chip segmentation: (**a**) Segmentation frequency for $V_c$ = 800 m/min, $f$ = 0.3 mm/rev; (**b**) Segmentation frequency for $V_c$ = 400 m/min, $f$ = 0.3 mm/rev; (**c**) Segmentation frequency for $V_c$ = 800 m/min, $f$ = 0.4 mm/rev; (**d**) Segmentation frequency for $V_c$ = 400 m/min, $f$ = 0.4 mm/rev; (**e**) Degree of chip segmentation for $V_c$ = 800 m/min, $f$ = 0.3 mm/rev; (**f**) Degree of chip segmentation for $V_c$ = 400 m/min, $f$ = 0.3 mm/rev; (**g**) Degree of chip segmentation for $V_c$ = 800 m/min, $f$ = 0.4 mm/rev; (**h**) Degree of chip segmentation for $V_c$ = 400 m/min, $f$ = 0.4 mm/rev.

Figure 7 shows a highly segmented chip morphology (with higher degree of chip segmentation) generated for $V_C$ = 800 m/min, $f$ = 0.4 mm/rev, $r_{eq}$ = 180 µm (level 23). In shear bands, the stiffness is fully degraded, with almost zero value for stresses. This shows the probability of fracture in the primary shear zone. Similar trends can also be seen in Figure 8 with variation of feed. The degree of chip segmentation is highly influenced by the change in feed, although by decreasing feed, segmentation frequency increases (due to length effect, longer segments of chips are generated), but this effect is not as pronounced as can be seen for the degree of segmentation.

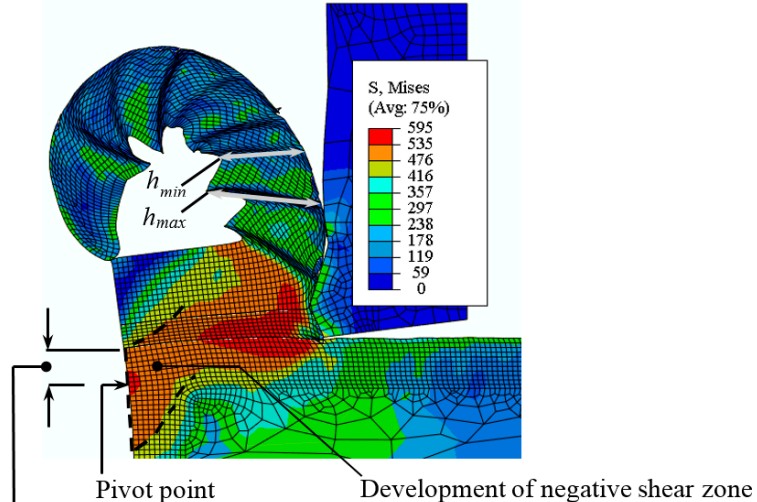

Pivot point distance from machined surface = 219.8 µm

**Figure 7.** Chip morphology for $V_C$ = 800 m/min, $f$ = 0.4 mm/rev, $r_{eq}$ = 180 µm (level 23).

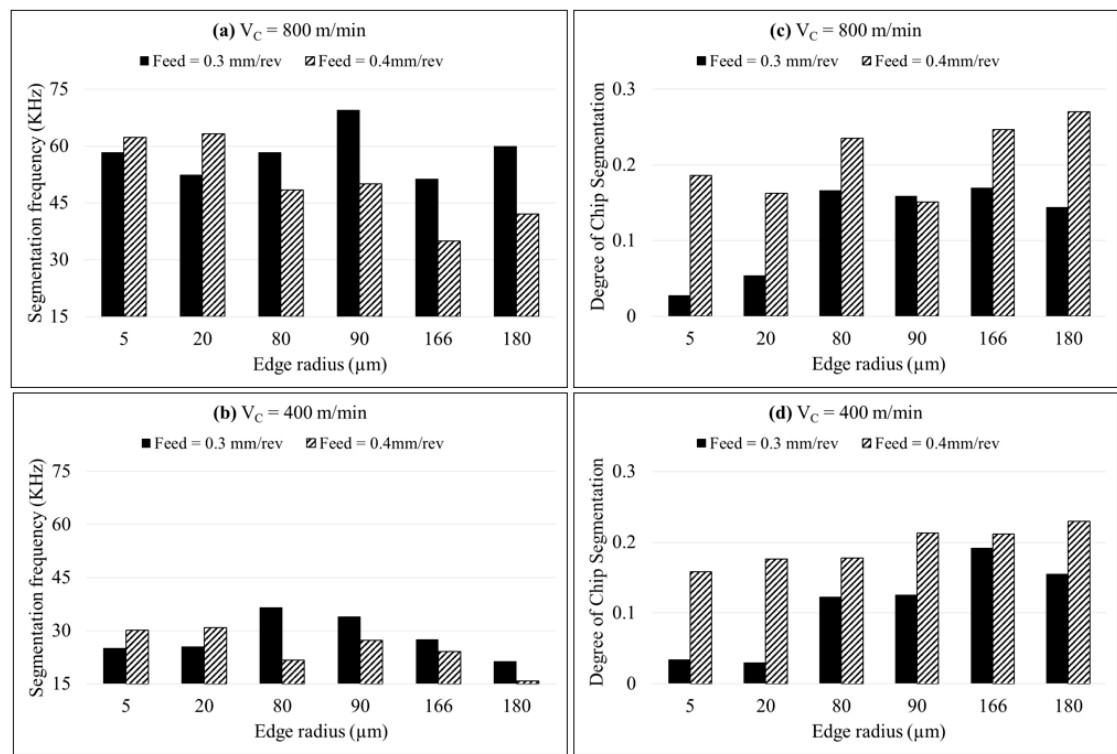

**Figure 8.** Feed rate effect on segmentation frequency and degree of chip segmentation: (**a**) Segmentation frequency for $V_c$ = 800 m/min; (**b**) Segmentation frequency for $V_c$ = 400 m/min; (**c**) Degree of chip segmentation for $V_c$ = 800 m/min; (**d**) Degree of chip segmentation for $V_c$ = 400 m/min.

Considering the above, it can be summarized that cutting speed greatly influences the chip segmentation frequency, while feed and tool edge radius largely effect the degree of chip segmentation. The thermal softening phenomenon plays a vital role in chip segmentation at higher cutting speeds, lower feed rates, and with smaller tool edge radius values (mainly in increasing segmentation frequency), while crack propagation in primary shear bands occurs at higher values of cutting edge radius and feed (largely influence segmentation degree). To predict optimal combinations of speed, feed, and tool edge radius to minimize the generation of segmented chip morphology (segmentation frequency and degree of chip segmentation), statistical analyses are performed in the next section.

### 3.1.2. Cutting Parameters and Tool Geometry Effects on End (Exit) Burr Formation

During the course of chip formation, as the tool keeps on advancing in the cutting direction towards the end of the workpiece, a negative shear zone starts to grow from the workpiece free end (exit end) towards the primary shear zone (Figures 3 and 7). The formation of the negative shear zone is specifically due to the bending load experienced by the workpiece free end during tool advancement in the cutting direction. As the tool advances further, the bending load keeps on increasing, the material experiences higher stresses in this deformation zone, and a pivot point (high stressed point) appears on the exit edge of the workpiece (Figures 3 and 7). The location of the "pivot point" is measured from the machined surface along the *y*-axis. The distance of the "pivot point" has a direct relationship with burr lengths (produced at the exit end)—longer distances represent longer burr lengths. The pivot point distance highly depends on the cutting parameters, materials, and tool geometry. During the course of cutting, the negative shear zone expands further around the pivot point and reaches the tool edge. Higher stresses far ahead of the tool tip position (due to the negative shear zone) promote the material's ductile failure and initiation of cracks in the chip separation zone far ahead of the tool tip (Figure 9). The material deviates from the actual cutting phenomenon, the chip formation process ceases, the tool pushes away the boot-type chip (combination of chip and uncut material), and the end burr (workpiece's deformed exit edge) appears at the end of the workpiece. Figure 9 shows early and advanced failure of chip separation zone material with formation of cracks and generation of an end burr for $V_C$ = 800 m/min, $f$ = 0.4 mm/rev, $r_{eq}$ = 180 μm (level 23).

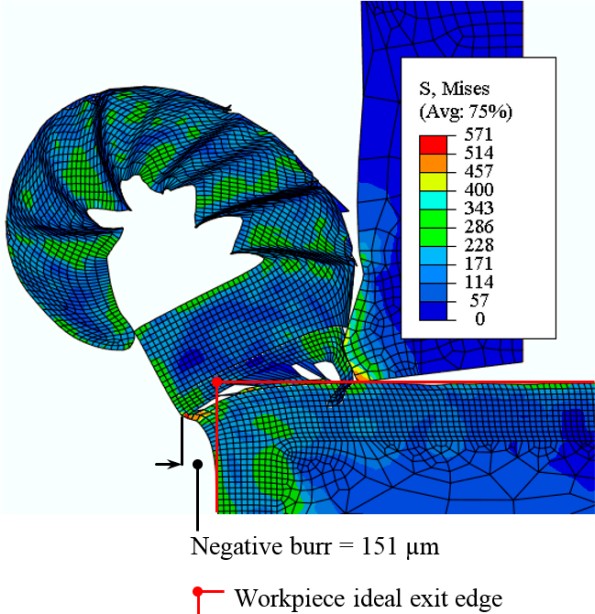

Negative burr = 151 μm

Workpiece ideal exit edge

**Figure 9.** The chip separation zone's material advance failure and formation of negative burr for $V_C$ = 800 m/min, $f$ = 0.4 mm/rev, $r_{eq}$ = 180 μm (level 23).

During machining of aluminum alloys, for various combinations of cutting parameters, both negative and positive burrs at the end of the workpiece have been reported in the literature [3]. Positive burrs (without considerable damage to workpiece edge) are normally generated at lower feed values, and vice versa [3]. In the present work, for AA2024-T351, with investigated combinations of cutting speed, feed, and tool edge geometry, only negative burrs (with edge breakout) were formed. It is found that machining performed with higher feeds along with larger tool edge radii produces highly stressed and more widened shear zones (both primary and negative), and the pivot point location is further away from the machined surface, generating longer burrs than for machining performed at lower feed rates and with smaller tool edge radii. Figures 10 and 11 quantify and produce a trend for exit burrs as a function of the feed and tool edge radius. On the other hand, speed variation was been found to have non-noticeable effects in changing exit burr lengths (Figure 12). The results, in general, are consistent with the findings of experimental burr formation studies performed on aluminum alloys [3,32]. Table 6 details numerically computed exit burr lengths for twenty-four various combinations (defined in Table 2) of cutting speed ($V_C$), feed ($f$), and tool edge geometries.

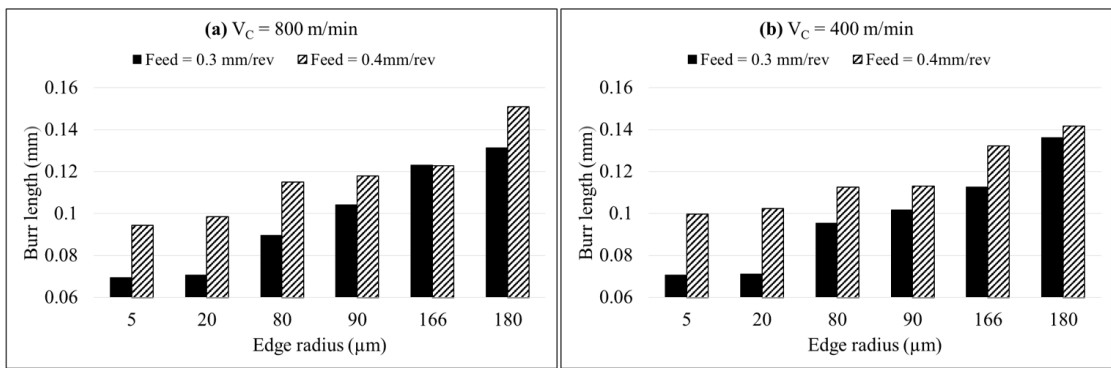

**Figure 10.** Feed rate variation effects on exit burr lengths: (**a**) Burr length at $V_c$ = 800 m/min; (**b**) Burr length at $V_c$ = 400 m/min.

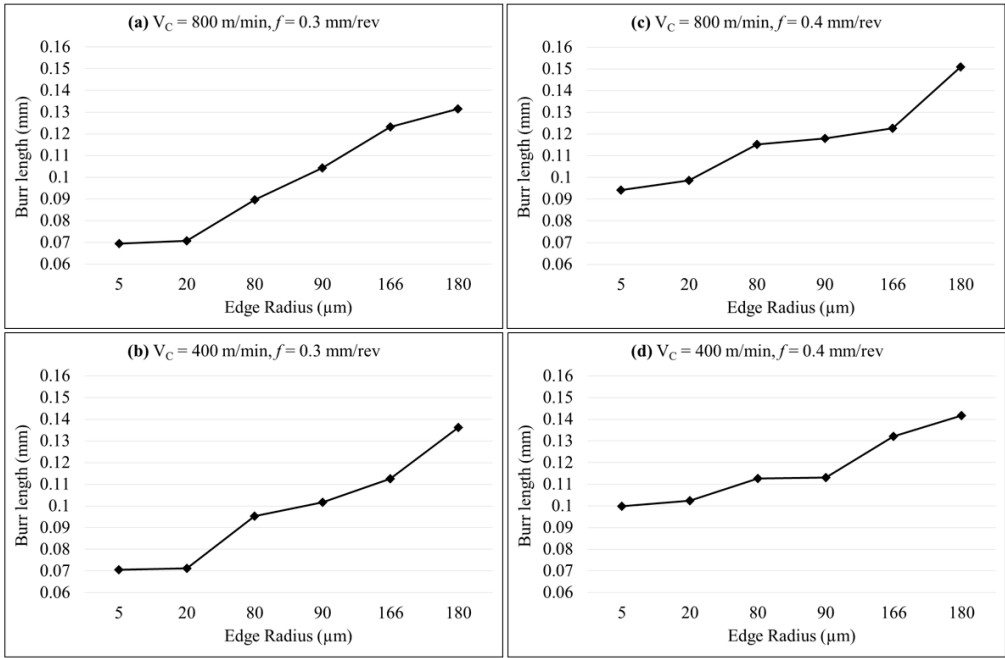

**Figure 11.** Edge radius variation effects on exit burr lengths: (**a**) Burr length for $V_c$ = 800 m/min, $f$ = 0.3 mm/rev; (**b**) Burr length for $V_c$ = 400 m/min, $f$ = 0.3 mm/rev; (**c**) Burr length for $V_c$ = 800 m/min, $f$ = 0.4 mm/rev; (**d**) Burr length for $V_c$ = 400 m/min, $f$ = 0.4 mm/rev.

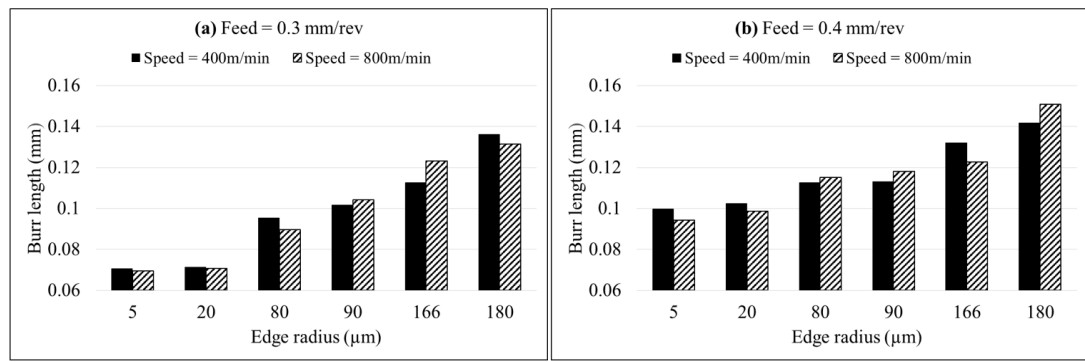

**Figure 12.** Cutting speed variation effects on exit burr lengths: (**a**) Burr length for $f$ = 0.3 mm/rev; (**b**) Burr length for $f$ = 0.4 mm/rev.

**Table 6.** Numerically computed exit burr lengths.

| Level | Factors | | | Numerically Computed Exit Burr Lengths (µm) |
|---|---|---|---|---|
| | Tool Edge Equivalent Radius, $r_{eq}$ (µm) | Cutting Speed, $V_C$ (m/min) | Feed Rate, $f$ (mm/rev) | |
| 1 | 5 | 800 | 0.3 | 69.5 |
| 2 | 5 | 400 | 0.3 | 70.6 |
| 3 | 20 | 800 | 0.3 | 70.9 |
| 4 | 20 | 400 | 0.3 | 71.2 |
| 5 | 80 | 800 | 0.3 | 89.6 |
| 6 | 80 | 400 | 0.3 | 95.4 |
| 7 | 90 | 800 | 0.3 | 104.3 |
| 8 | 90 | 400 | 0.3 | 101.8 |
| 9 | 166 | 800 | 0.3 | 123.2 |
| 10 | 166 | 400 | 0.3 | 112.6 |
| 11 | 180 | 800 | 0.3 | 131.5 |
| 12 | 180 | 400 | 0.3 | 136.2 |
| 13 | 5 | 800 | 0.4 | 94.3 |
| 14 | 5 | 400 | 0.4 | 99.8 |
| 15 | 20 | 800 | 0.4 | 98.6 |
| 16 | 20 | 400 | 0.4 | 102.3 |
| 17 | 80 | 800 | 0.4 | 115.2 |
| 18 | 80 | 400 | 0.4 | 112.6 |
| 19 | 90 | 800 | 0.4 | 118 |
| 20 | 90 | 400 | 0.4 | 113 |
| 21 | 166 | 800 | 0.4 | 122.7 |
| 22 | 166 | 400 | 0.4 | 132.1 |
| 23 | 180 | 800 | 0.4 | 150.9 |
| 24 | 180 | 400 | 0.4 | 141.8 |

### 3.2. Statistical Analysis and Optimization

In the preceding section, finite element method (FEM) approach was employed to predict the likelihood of chip segmentation features (segmentation frequency and degree of chip segmentation) and exit burr formation under various combinations of speed, feed, and tool edge radius. Various associated phenomena such as maximum nodal temperature, material stiffness degradation, early fracture of material in the tool's advancement direction, and location of the pivot point are also discussed. Interesting conclusions can be drawn for optimizing the machining of AA2024-T351 using tungsten carbide inserts. Nevertheless, further investigations are required to predict optimum combinations of speed, feed, and tool edge radius to minimize the generation of segmented chip morphology (segmentation frequency and degree of chip segmentation) and reduce burr formation. The relative

significance of each cutting parameter on the latter phenomenon would also be interesting from a production engineer's perspective. Predictive models of chip segmentation features (segmentation frequency and degree of chip segmentation) and exit burr lengths would be advantageous to minimize the cutting trials to optimize the cutting. In this framework, the present section exploits statistical analysis tools, such as Taguchi's design of experiment (DOE), analysis of variance (ANOVA), and response surface methodology (RSM).

### 3.2.1. Statistical Analyses on Burr Optimization

To determine the optimum combination of cutting parameters (speed, feed, and tool edge radius) for minimum end burr lengths, Taguchi's DOE is employed. The quality criterion approach "the-smaller-the-better" is used for the data (exit burr lengths computed over twenty-four tests via finite element analysis (FEA) and Equation (12) is used to determine the signal-to-noise (S/N) ratio.

$$\frac{S}{N} = -10log\left(\sum\left(\frac{y_i^2}{n}\right)\right) \tag{12}$$

In the relationship, "$y_i$" represents the response value of the $i$th test and "$n$" is the number of test repetitions (taken as one). Feed and speed have two levels of variations ($f = 0.3$ and $0.4$ mm/rev and $V_C = 800$ and $400$ m/min), while tool edge radius has six levels of variations ($r_{eq} = 5, 20, 80, 90, 166,$ and $180$ μm). The parametric combination $V_C = 800$ m/min, $f = 0.3$ mm/rev, $r_{eq} = 5$ μm (Level 1, Table 2) represents the optimum combination for generation of minimum burr, as can be figured out by the plots of main effects of S/N ratio (Figure 13a) and data mean (Figure 13b). Table 7 results show that the edge radius is the most influential factor and speed is the least influential factor in burr formation. Results show a good match with the experimental findings of Niknam and Songmene [32].

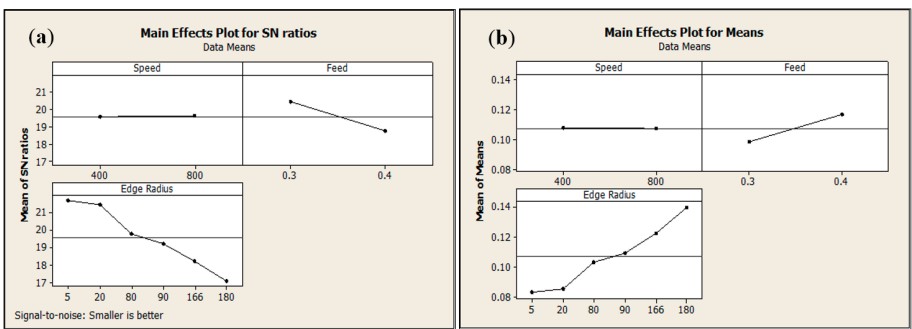

**Figure 13.** Exit burr lengths as a function of cutting parameters: (**a**) variation of signal to noise (S/N) ratios and (**b**) data means.

**Table 7.** Response table for data means.

| Level | Factor (Speed) | Factor (Feed) | Factor (Edge Radius) |
|---|---|---|---|
| 1 | 0.107385925 | 0.098054667 | 0.083556375 |
| 2 | 0.107456283 | 0.116787542 | 0.08575725 |
| 3 | - | - | 0.10319225 |
| 4 | - | - | 0.1092785 |
| 5 | - | - | 0.122657 |
| 6 | - | - | 0.14008525 |
| Difference | $7.03583 \times 10^{-5}$ | 0.018732875 | 0.056528875 |
| Rank | 3 | 2 | 1 |

Next, to establish a relationship between exit burr lengths and machining parameters, a second order multiple regression model (Equation (13)) based on RSM is used. The developed regression model (Equation (14) using Minitab software (Minitab, 16.2, Minitab-LLC, State College, PA, USA,

2010). The predicted value for burr length (for optimal cutting parameters: $V_C = 800$ m/min, $f = 0.3$ mm/rev, $r_{eq} = 5$ µm) to generate minimum burr using Equation (14) matches the value acquired through finite element simulation (Table 6).

$$y = \beta_0 + \sum_{i=1}^{3} \beta_i x_i + \sum_{i=1}^{3} \beta_{ij} x_i x_j + \sum_{i=1}^{3} \beta_{ii} x_i^2 + \varepsilon \tag{13}$$

$$\begin{aligned} Burr\ length &= -0.0160032 - 1.41323e^{-6}(Speed) + 0.290049\ (Feed) - 0.000581417\ (Edge\ radius) \\ &-1.34362e^{-5}(Speed \times Feed) + 6.58783e^{-8}(Speed \times Edge\ radius) - 0.00104982\ (Feed \times Edge\ radius) \\ &+2.17843e^{-7}(Edge\ radius \times Edge\ radius) \end{aligned} \tag{14}$$

In Equation (14), quadratic terms of speed and feed are been included as they are insignificant. Finally, to determine the significance of the regression model and relative contribution of each of the machining parameters, analysis of variance (ANOVA) is performed. Terms used in ANOVA Table 8 are defined in Equations (15)–(18).

$$Sum\ of\ Squares\ (SS) = \frac{N}{nf} \sum_{i=1}^{nf} (\overline{y}_i - \overline{y})^2 \tag{15}$$

where $N$ is the total number of tests, $nf$ represents the level of each factor, $\overline{y}$ is the mean of the response, and $\overline{y}_i$ is the mean of the response at each level of the respective factor.

$$Mean\ square(Variance) : MS_i = \frac{SS}{DF_i} \tag{16}$$

$$Fisher\ Coefficient(F\text{-}value) = \frac{MS_i}{MS_{Error}} \tag{17}$$

$$Percent\ Contribution : PP(\%) = \left(\frac{SS}{SS_T}\right) \times 100 \tag{18}$$

**Table 8.** Analysis of variance (ANOVA) results for exit (end) burr.

| Source | DF | SS | MS | *F*-Value | *P*-Value | PP% | Remarks |
|---|---|---|---|---|---|---|---|
| Regression | 7 | 0.011474 | 0.001639 | 41.56 | 0 | - | Significant |
| Speed | 1 | 0 | 0 | 0.001 | 0.997 | 0 | Insignificant |
| Feed | 1 | 0.002106 | 0.002048 | 51.94 | 0.000002 | 17.39776952 | Significant |
| Edge Radius | 1 | 0.009051 | 0.00898 | 227.7 | 0 | 74.77075589 | Significant |
| Edge Radius × Edge Radius | 1 | 0.000011 | 0.000011 | 0.29 | 0.598 | 0.090871541 | Insignificant |
| Speed × Feed | 1 | 0 | 0 | 0.01 | 0.918 | 0 | Insignificant |
| Speed × Edge Radius | 1 | 0.000018 | 0.000018 | 0.46 | 0.508 | 0.148698885 | Insignificant |
| Feed × Edge Radius | 1 | 0.000288 | 0.000288 | 7.29 | 0.016 | 2.379182156 | Significant |
| Error | 16 | 0.000631 | 0.000039 | - | - | 5.212722016 | - |
| Total | 23 | 0.012105 | - | - | - | 100 | - |

In the ANOVA table, significance or insignificance is attributed to each of the source factors based on the Fisher coefficient value (*F*-value). ANOVA for significance level = 5% (95% confidence level) was performed. The probability values (*P*-values) of the regression model, feed, and edge radius are <0.05. This shows the significance of the regression model and the factors that contribute the most: feed and edge radius. Speed, "quadratic terms", and "interactive terms" have the least effect on burr formation. Table 8 also shows that the edge radius has the highest contribution in producing burr at 74.77%, the feed contribution is 17.39%, while speed variation has the least effect in exit burr formation. This hierarchy of contribution also confirms the findings of Taguchi's DOE methodology (Table 7).

It is interesting to note that ANOVA produced for "pivot point location" (considering it as target function, Table 9) has similar trends in term of % contribution of machining parameters in producing burr (Table 8). This helps to conclude that a distant pivot point location (for larger edge radius and higher feed values) is a strong sign that longer burr will be produced.

**Table 9.** ANOVA results for pivot point location.

| Source | DF | SS | MS | *F*-Value | *P*-Value | PP % | Remarks |
|---|---|---|---|---|---|---|---|
| Regression | 7 | 0.049389 | 0.007056 | 12.91 | 0 | - | Significant |
| Speed | 1 | 0.00037 | 0.000374 | 0.68 | 0.42 | 0.636493437 | Significant |
| Feed | 1 | 0.004971 | 0.004976 | 9.11 | 0.008 | 8.551375342 | Insignificant |
| Edge Radius | 1 | 0.04362 | 0.043299 | 79.25 | 0 | 75.03741549 | Insignificant |
| Edge Radius × Edge Radius | 1 | 0.000044 | 0.000044 | 0.08 | 0.781 | 0.075691111 | Insignificant |
| Speed × Feed | 1 | 0.00037 | 0.00037 | 0.68 | 0.423 | 0.636493437 | Insignificant |
| Speed × Edge Radius | 1 | 0.00001 | 0.00001 | 0.02 | 0.894 | 0.017202525 | Insignificant |
| Feed × Edge Radius | 1 | 0.000005 | 0.000005 | 0.01 | 0.924 | 0.008601263 | Insignificant |
| Error | 16 | 0.008742 | 0.000546 | - | - | 15.03844764 | - |
| Total | 23 | 0.058131 | - | - | - | 100 | - |

### 3.2.2. Statistical Analyses of Segmented Chip Morphology (Segmentation Frequency and Degree of Chip Segmentation)

Figure 14a,b presents the plots of the main effects of S/N ratios and data means on segmentation frequency, respectively. Analyses of plots show that segmentation frequency increases as speed increases, while higher feed and larger edge radii suppress the segmentation phenomenon, though their effect is negligible. The parametric combination $V_C$ = 400 m/min, $f$ = 0.4 mm/rev, $r_{eq}$ = 180 μm (level 24, Table 2) represents the optimum combination for generating the least amount of segmentation frequency.

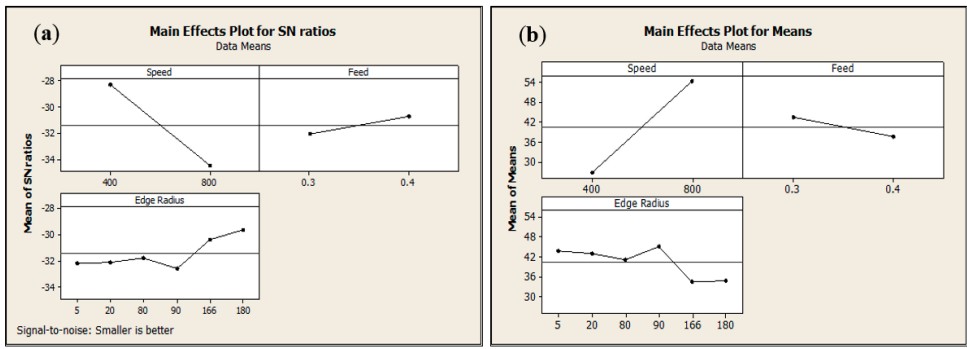

**Figure 14.** Segmentation frequency as a function of cutting parameters: (**a**) variation of signal-to-noise (S/N) ratios and (**b**) data means.

A second order multiple regression model based on RSM is presented in Equation (19) to define the relationship of segmentation frequency as a function of cutting parameters. In the model, quadratic terms of speed and feed are not included as they are insignificant.

$$
\begin{aligned}
\textit{Segmentation Frequency} = &-39.8568 + 0.11879\,(\textit{Speed}) + 105.777\,(\textit{Feed}) + 0.44496\,(\textit{Edge radius}) \\
&-0.120458\,(\textit{Speed} \times \textit{Feed}) - 8.47311e^{-5}(\textit{Speed} \times \textit{Edge radius}) - 1.00696\,(\textit{Feed} \times \textit{Edge radius}) \\
&-5.16853e^{-4}(\textit{Edge radius} \times \textit{Edge radius})
\end{aligned}
\tag{19}
$$

To outline the significance of the model and relative contribution of each of the cutting parameters on segmentation frequency, analysis of variance (ANOVA) is performed and results are summarized in Table 10. Results show that speed has the highest contribution in producing segmentation frequency at 76.63%, edge radius contributes 5.37%, while feed variation has the least effect in generating chips with high segmentation frequencies.

**Table 10.** ANOVA results for segmentation frequency.

| Source | DF | SS | MS | F Value | P-Value | PP % | Remarks |
|---|---|---|---|---|---|---|---|
| Regression | 7 | 5480.35 | 782.91 | 25.99 | 0 | - | Significant |
| Speed | 1 | 4569.18 | 4537.35 | 150.65 | 0 | 76.6351629 | Significant |
| Feed | 1 | 196.94 | 213.16 | 7.08 | 0.017 | 3.303115435 | Insignificant |
| Edge Radius | 1 | 320.62 | 302.58 | 10.05 | 0.006 | 5.377500105 | Insignificant |
| Edge Radius × Edge Radius | 1 | 64.15 | 64.15 | 2.13 | 0.164 | 1.075936098 | Insignificant |
| Speed × Feed | 1 | 34.82 | 34.82 | 1.16 | 0.298 | 0.584007715 | Insignificant |
| Speed × Edge Radius | 1 | 29.98 | 29.98 | 1 | 0.333 | 0.502830307 | Insignificant |
| Feed × Edge Radius | 1 | 264.65 | 264.65 | 8.79 | 0.009 | 4.438760535 | Insignificant |
| Error | 16 | 481.92 | 30.12 | - | - | 8.082854627 | - |
| Total | 23 | 5962.25 | - | - | - | 100 | - |

As discussed in Section 3.1.1, machining performed at higher speeds generates higher cutting temperatures (Figure 3), leading to thermal softening and generation of adiabatic shear bands (segmented chips). In this context, ANOVA analysis is performed for "maximum nodal temperature" (considering it as target function, Table 11) to figure out the % contribution of machining parameters (speed, feed, and tool edge radius) in influencing the temperature rise. It can be seen (Table 11) that edge radius has the highest contribution to temperature variation; indeed, temperature decreases as the edge radius increases (Figure 4), whereas speed is the second highest contributor in effecting the temperature; temperature increases as speed increases (Figure 4). Feed has been found to have the least effect on maximum temperature variations. Further analyses of Tables 10 and 11 help to conclude that higher temperatures produced at higher cutting speeds promote thermal softening and generation of more frequent adiabatic shear bands (higher segmentation frequency), whereas higher feed and larger edge radii reduce segmentation frequencies, though their effects are minimal.

**Table 11.** ANOVA results for maximum nodal temperature.

| Source | DF | SS | MS | F Value | P-Value | PP % | Remarks |
|---|---|---|---|---|---|---|---|
| Regression | 7 | 108782 | 15540.3 | 12.2 | 0 | - | Significant |
| Speed | 1 | 29698 | 28782.6 | 22.59 | 0 | 22.99140667 | Significant |
| Feed | 1 | 992 | 882 | 0.69 | 0.418 | 0.767980181 | Insignificant |
| Edge Radius | 1 | 68989 | 68900.8 | 54.07 | 0 | 53.4094604 | Insignificant |
| Edge Radius × Edge Radius | 1 | 25 | 25.5 | 0.02 | 0.889 | 0.019354339 | Insignificant |
| Speed × Feed | 1 | 1247 | 1246.9 | 0.98 | 0.337 | 0.965394441 | Insignificant |
| Speed × Edge Radius | 1 | 5282 | 5282.4 | 4.15 | 0.059 | 4.089184795 | Insignificant |
| Feed × Edge Radius | 1 | 2548 | 2547.6 | 2 | 0.177 | 1.972594256 | Insignificant |
| Error | 16 | 20387 | 1274.2 | - | - | 15.78307657 | - |
| Total | 23 | 129170 | - | - | - | 100 | - |

Figure 15a,b presents the plots of the main effects of S/N ratios and data means on "degree of chip segmentation", respectively. Analyses of plots show that all cutting parameters promote segmentation degree, though speed's effect seems negligible. The parametric combination $V_C = 400$ m/min, $f = 0.3$ mm/rev, $r_{eq} = 5$ μm (level 2, Table 2) represents the optimum combination for generation of chips with the least degree of segmentation.

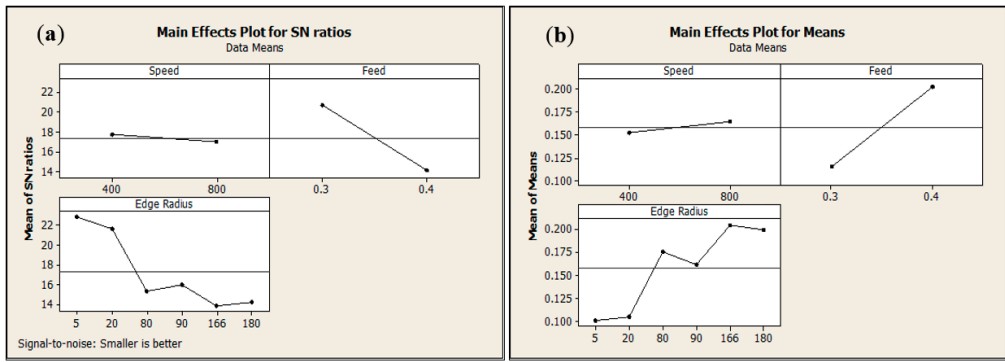

**Figure 15.** Degree of chip segmentation as a function of cutting parameters: variation of signal to noise (S/N) ratios (**a**) and (**b**) data means.

A second order multiple regression model based on RSM is presented in Equation (20) to define the relationship of the degree of chip segmentation as a function of cutting parameters. In the model, quadratic terms of speed and feed are been included as they are insignificant.

$$
\begin{aligned}
Degree\ of\ Segmentation = & -0.317506 - 5.7063\ (Speed) + 1.1144\ (Feed) + 0.22237096\ (Edge\ radius) \\
& + 0.000102409\ (Speed \times Feed) - 1.36297e^{-9}(Speed \times Edge\ radius) - 0.00342995\ (Feed \times Edge\ radius) \\
& - 3.03896e^{-6}(Edge\ radius \times Edge\ radius)
\end{aligned}
\tag{20}
$$

To outline the significance of the model and relative contribution of each of the cutting parameters on the degree of chip segmentation, analysis of variance (ANOVA) is performed and results are summarized in Table 12. Results show that speed has the least contribution to producing highly segmented chips (with high degree of chip segmentation), while feed (43.895%) and edge radii (36.46%) significantly affect the production of highly segmented chips. Finite element analyses provide explicit explanation in this context (Section 3.1.1). Larger material area experiences severe plastic deformation when cutting is performed at higher feed rates supplemented with larger tool radii. Material stiffness degrades, leading to crack initiation and propagation in primary shear bands, resulting in highly segmented chips.

**Table 12.** ANOVA results for degree of chip segmentation.

| Source | DF | SS | MS | *F* Value | *P*-Value | PP % | Remarks |
|--------|----|-----|-----|-----------|-----------|------|---------|
| Regression | 7 | 0.08866 | 0.012666 | 14.49 | 0 | - | Significant |
| Speed | 1 | 0.000865 | 0.000864 | 0.99 | 0.335 | 0.842710312 | Significant |
| Feed | 1 | 0.045057 | 0.044174 | 50.54 | 0 | 43.89595207 | Insignificant |
| Edge Radius | 1 | 0.037425 | 0.038387 | 43.92 | 0 | 36.46061669 | Insignificant |
| Edge Radius × Edge Radius | 1 | 0.002218 | 0.002218 | 2.54 | 0.131 | 2.160845633 | Insignificant |
| Speed × Feed | 1 | 0.000025 | 0.000025 | 0.03 | 0.867 | 0.024355789 | Insignificant |
| Speed × Edge Radius | 1 | 0 | 0 | 0 | 0.998 | 0 | Insignificant |
| Feed × Edge Radius | 1 | 0.003071 | 0.003071 | 3.51 | 0.079 | 2.991865166 | Insignificant |
| Error | 16 | 0.013984 | 0.000874 | - | - | 13.62365434 | - |
| Total | 23 | 0.102645 | - | - | - | 100 | - |

## 4. Conclusions

The paper provides a staggered comprehension-to-optimization approach for chip segmentation and end burr (exit burr) formation phenomena in machining of an aerospace-grade aluminum alloy AA2024-T351. These phenomena effect tool life, workpiece machined surface quality and integrity, and hence the overall productivity. Primarily, a finite-element-based cutting model has been established and used to simulate orthogonal machining and chip formation processes for multiple parametric combinations of cutting speed, feed, and tool edge geometry. Results concerning chip

segmentation (segmentation frequency and degree of segmentation) and end burr are numerically computed and comprehensively analyzed. To validate the numerical machining model, cutting forces, chip segmentation frequency, and chip morphology results are adequately compared with their experimental counterparts. Then, statistical optimization techniques such as Taguchi's DOE and ANOVA are employed to identify optimum cutting parameters and their % influence in effecting chip segmentation and end burr formation processes. Lastly, RSM-based quadratic predictive models for the aforementioned phenomena are presented.

The results presented in the current work are equally interesting for designers and researchers, providing further insight into machining and related phenomena. From a production engineering perspective, they provide optimum cutting conditions to enhance productivity through optimum selection of tool geometry and cutting parameters. Important findings of the present work are listed below.

- Machining operations performed with chamfer (T-land) edges can be represented with equivalent hone (round) edge radii.
- Only negative burr with a boot-type chip was witnessed for all investigated parametric cutting combinations of speed, feed, and tool edge geometry in machining of AA2024-T351.
- The negative shear zone is wider for cutting performed at higher cutting feed accompanied with larger tool edge radii. This promotes the material's early ductile failure, initiation, and progression of fracture in the chip separation zone far ahead of the tool tip location. Consequently, the material escapes the cutting process and the tool pushes away the boot-type chip (combination of chip and uncut workpiece material), and a longer negative end burr (deformed workpiece exit edge) appears at the exit edge of the workpiece. Statistical analyses show that tool edge radius is the major contributor (74%), while feed rate contributes up to 17.4% in generating burr. Cutting speed variation has been found to have negligible effects on burr quantification.
- Pivot point (the highly stressed point in the negative shear zone) location on the exit edge of the workpiece shows a direct relation in quantifying burr lengths. The distant location of the pivot point from the machined surface results in longer burr lengths, and vice versa.
- Higher cutting speeds enhance thermal softening and more frequent generation of chip shear bands (high frequency of chip segmentation). Finite-element-based parametric analyses and subsequent application of statistical optimization approaches show that speed is the highest contributor (76%) among the cutting parameters in generating highly segmented chips, significantly more so than feed and tool edge radii. Any variation in the latter parameters were found to have insignificant effects in this area.
- Wider workpiece materials undergo severe plastic deformation when machining is performed at higher cutting feeds complemented with larger tool edge radii. The material stiffness degrades easily, leading to crack initiation and propagation in primary and secondary shear zones, resulting in highly segmented chips (chips with a higher degree of chip segmentation). However, cutting speeds, on the other hand, did not been noticeably effect the degree of chip segmentation. Statistical analyses show that feed and tool edge radius both dominantly effect the phenomena, with contributions of 43.9% and 36.4%, respectively.
- Optimum cutting parametric combinations of feed, speed, and tool edge radius to minimize chip segmentation and exit burr formation have been presented. Furthermore, quadratic regression models have been proposed to quantify segmentation frequencies, degree of segmentation, and exit burr lengths as functions of cutting speed, feed, and tool edge radius.

In future studies, a more realistic friction model along with the most accurate heat fraction coefficient, *J*, will be incorporated into the finite element model to present more realistic results of industrial interest. Furthermore, the study will be extended for other materials and processes, such as drilling.

**Funding:** This research received no external funding.

**Acknowledgments:** Technical support provided by Francois Girardin of Laboratoire Vibrations Acoustique, INSA de Lyon, France is highly appreciated.

**Conflicts of Interest:** The authors declare no conflict of interest.

## Abbreviations

| | |
|---|---|
| $A$ | Initial yield stress (MPa) |
| $a_P$ | Cutting depth (mm) |
| $B$ | Hardening modulus (MPa) |
| $C$ | Strain rate dependency coefficient |
| $Cp$ | Specific heat (Jkg$^{-1}$ °C$^{-1}$) |
| $D$ | Damage evolution parameter |
| $D1\ D5$ | Coefficients of Johnson–Cook material shear failure initiation criterion |
| $E$ | Young's modulus (MPa) |
| $f$ | Feed rate (mm/rev) |
| $G_f$ | Fracture energy (N/m) |
| $KC_{I, II}$ | Fracture toughness ($MPa\ \sqrt{m}$) for failure mode I and mode II |
| $l_\beta$ | Chamfer length |
| $m$ | Thermal softening coefficient |
| $n$ | Work-hardening exponent |
| $P$ | Hydrostatic pressure (MPa) |
| $\dot{q}_p$ | Heat generation rate due to plastic deformation W/m$^3$ |
| $\dot{q}_f$ | Heat generation rate due to friction W/m$^3$ |
| $r_\beta$ | Hone edge radius |
| $r_{eq}$ | Equivalent edge radius |
| $T$ | Temperature at a given calculation instant (°C) |
| $T_m$ | Melting temperature (°C) |
| $T_r$ | Room temperature (°C) |
| $\bar{u}$ | Equivalent plastic displacement (mm) |
| $\bar{u}_f$ | Equivalent plastic displacement at failure (mm) |
| $V_C$ | Cutting speed (m/min) |
| $\bar{\sigma}$ | Stress, MPa |
| $\bar{\sigma}_{JC}$ | Johnson-Cook equivalent stress (MPa) |
| $\sigma_y$ | Yield stress (MPa) |
| $T_f$ | Frictional shear stress, MPa |
| $\frac{P}{\bar{\sigma}}$ | Stress triaxiality |
| $\bar{\varepsilon}$ | Equivalent plastic strain |
| $\dot{\bar{\varepsilon}}$ | Plastic strain rate (s$^{-1}$) |
| $\dot{\bar{\varepsilon}}_0$ | Reference strain rate (10$^{-3}$ s$^{-1}$) |
| $\bar{\varepsilon}_f$ | Equivalent plastic strain at failure |
| $\Delta\bar{\varepsilon}$ | Equivalent plastic strain increment |
| $\bar{\varepsilon}_{0i}$ | Plastic strain at damage initiation |
| $\eta_p$ | Inelastic heat fraction |
| $\eta_f$ | Frictional work conversion factor |
| $\omega$ | Damage initiation criterion |
| $\nu$ | Poisson's ratio |
| $\alpha_d$ | Expansion coefficient (μm m$^{-1}$ °C$^{-1}$) |
| $\lambda$ | Thermal conductivity (W m$^{-1}$ C$^{-1}$) |
| $\rho$ | Density (kg/m$^3$) |
| $\gamma_o$ | Rake angle (degrees) |

| | |
|---|---|
| $\gamma_\beta$ | Chamfer angle (degrees) |
| $\beta_0$ | Constant of regression model equation |
| $\beta_i$ and $\beta_j$ | Coefficient of linear terms of regression model equation |
| $\beta_{ij}$ | Coefficient of quadratic terms of regression model equation |
| $x_i$ and $x_j$ | Explicative variables (control factors) |
| $\varepsilon$ | Random error |
| *ANOVA* | Analysis of variance |
| *DOE* | Design of experiment |
| *RSM* | Response surface methodology |
| *DF* | Degrees of freedom |
| *MS* | Mean squares (variance) |
| *SS* | Sum of squares |
| *S/N ratio* | Signal-to-noise ratio |
| *PP* | Percent contribution |
| *P-value* | Probability of significance |
| *F-value* | Fisher coefficient (variance ratio) |

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
