# Peer review of "Effects of Tool Edge Geometry on Chip Segmentation and Exit Burr: A Finite Element Approach"

_metals, doi:10.3390/met9111234_

Round 1

Reviewer 1 Report

The work offers a good attempt to study the effects of tool edge geometry on quantification of exit burr and chip segmentation using FE analysis. 

More recent references in modeling cutting processes using FE analysis should be added, FOR EXAMPLE:

- Hegab, H., Kishawy, H. A., Umer, U., & Mohany, A. (2019). A model for machining with nano-additives based minimum quantity lubrication. The International Journal of Advanced Manufacturing Technology, 1-16.

Author Response

I would like to thank the reviewer for his constructive comments and valuable suggestions. These comments not only encouraged but also guided me to improve the quality of the work. I have followed the instructions carefully and to the best of my understanding. All the additions to the original manuscript are highlighted in red. Rephrasing and rewording are done to improve the logical flow. Hope the revised manuscript will meet the required standard for approval. Please find the point-by-point responses to the reviewers’ comments as:

Comment: English language and style

( ) Extensive editing of English language and style required 
(x) Moderate English changes required 
( ) English language and style are fine/minor spell check required 
( ) I don't feel qualified to judge about the English Language and Style

Response: I have tried my best to refine the level of English to meet the required standard. Complex sentences are simplified and explained for better comprehension in the revised manuscript.

Comment: More recent references in modeling cutting processes using FE analysis should be added, FOR EXAMPLE: “ Hegab, H., Kishawy, H. A., Umer, U., & Mohany, A. (2019). A model for machining with nano-additives based minimum quantity lubrication. The International Journal of Advanced Manufacturing Technology, 1-16”

Response: Reference [16] and related details (Line 48-52) are now included in the revised manuscript.

Reviewer 2 Report

The scientific work is very good. It has a topic of current interest, the research is well done and the presentation is very clear.

I have only one small observation. Please complete the material standard on line 30.

Author Response

I would like to thank the reviewer for his constructive comments and valuable suggestions. These comments not only encouraged but also guided me to improve the quality of the work. I have followed the instructions carefully and to the best of my understanding. All the additions to the original manuscript are highlighted in red. Rephrasing and rewording are done to improve the logical flow. Hope the revised manuscript will meet the required standard for approval. Please find the point-by-point responses to the reviewers’ comments as:

Comment: English language and style

( ) Extensive editing of English language and style required 
( ) Moderate English changes required 
(x) English language and style are fine/minor spell check required 
( ) I don't feel qualified to judge about the English Language and Style

Response: I have tried my best to refine the level of English to meet the required standard. Complex sentences are simplified and explained for better comprehension in the revised manuscript.

Comment: I have only one small observation. Please complete the material standard on line 30.

Response: Complete details of material standards is now included in the revised manuscript (Line 30) and elsewhere in the revised manusript.

Reviewer 3 Report

Originality

In drilling of metallic materials or composite/metal stacks, chip segmentation and exit burr are two main problems since they can damage the hole surface during chip evacuation and cause incorrect positioning of the connecting elements (rivet or screw ...). This frequently leads to repairs and incurs additional costs. The control of these two phenomena is therefore of paramount importance for certain sectors such as aeronautics. Thereby, the article is sufficiently novel and interesting to warrant publication. The integrity of material during drilling is a current challenge. Few works have been dedicated to the relation of tool edge geometry with exit burr and chip segmentation. This work is then of interest for the machining community in order to improve the knowledge in drilling of metallic materials. The results can be also extended to the drilling of composite/metal stacks, very common in the aeronautic sector. In that sense the work adds to the canon of knowledge.

The article is between the top 30% of papers in this field. It is an interesting contribution.

The article adheres to the journal's standards.

Structure

Title

The title clearly describes the article, but is long enough.

Abstract

The abstract reflects well enough the content of the article.

Introduction

The introduction describes what the author hoped to achieve accurately, and clearly states the problem being investigated.

The authors summarize relevant research to provide context, and explain what other authors findings are being challenged or extended in the case of exit burr and chip segmentation. The present work aims to examine the effect of tool edge geometry -hone (round) edge, Chamfer edge- also called “tool edge preparation” on chip formation, chip segmentation frequency, degree of chip segmentation and exit burr formation processes.

Please add a paragraph explaining how the tool edge geometry affects chip segmentation and exit burr since it is the core of the paper. Define the parameters concerning edge preparation which will be studied after (see paper of Denkena et al. DOI: 10.1016/j.procir.2012.04.033).

Figures and Tables

The figures and tables inform quite well the reader. They are of good quality and do not need any modification. Just the following remarks:

Table 5, which gives the results of cutting forces and chip segmentation frequencies, needs to be completed giving the results for the 24 DoE lines that have been experimentally performed. These data could be interesting for other authors. In figure 6, add a picture of the experimentally obtained chip as in figure 2. Idem for figure 8. Add a table with the dimensions of the exit burr as table 5 for cutting forces and chip segmentation frequency.

FEM modelling and Results

Another aim of the presented work is to provide further insight in chip and burr formation in machining of AA2024-T351 and optimize cutting parameters and tool edge design for improved productivity employing FE based design and analysis approach. Numerically computed results of chip morphology, cutting forces and chip segmentation frequency are compared with the ones obtained previously by performing orthogonal cutting experimental investigations on AA2024-T351 under similar cutting conditions.

The geometrical model, the meshing, the constraints and hypothesis are well defined and allow to repeat the simulation without problems.

Mesh density of the order of 20 μm is chosen in “chip separation zone”, chip and upper layer (~ 0.3 mm) of machined workpiece. Why a finer mesh has not been chosen in the chip separation zone (since this zone may be very small of the order of 5 microns)?

The choices of the material behaviour, the chip separation damage law and the thermal models are justified and well documented.

The choice of the coefficient of distribution of the friction energy between the chip and the tool (J = 0.5) is questionable and does not take into account the thermal properties of the two materials (aluminium alloy and tool). The use of Coulomb`s friction law is also a simplistic model. Why not a more realistic model (see DOI 10.1007/s00170-016-9149-4)

The results are interesting since the basic phenomenon controlling chip segmentation is the presence of adiabatic shear zones. Cutting speed influences greatly the chip segmentation frequency, while feed and tool edge radius affects largely the degree of chip segmentation. Thermal softening phenomenon plays vital role in chip segmentation at higher cutting speeds, lower feed and smaller tool edge radius values (mainly in increasing segmentation frequency), while crack propagation in primary shear bands are occurred at higher values of cutting edge radius and feed (largely influence segmentation degree). These results are in agreement with other works.

Experimental Method and Results

Parametric analyses using various combinations of cutting speed, feed and tool edge geometry on later processes for orthogonal cutting of AA2024-T351 are presented. A full factorial Taguchi`s design of experiment (DOE) technique are employed to determine optimum combinations of tool edge geometry, cutting speed and feed to curtail burr lengths, chip segmentation frequency and degree of chip segmentation. Analysis of variance (ANOVA) is performed to determine percentage influence of these factors on exit burr lengths, segmentation frequency and degree of segmentation. Response surface methodology (RSM) based quadratic predictive models are also proposed.

The paragraph concerning the experimental procedures does not accurately explain how the data were collected. There is no sufficient information present to replicate the research. The article does not identify the procedures followed; the equipment and materials which have been used.

Are the uncoated carbide inserts CCGX 12 04 08-AL 93 H10, the ones used in the experimental part? How has been modified the cutting edge? How many times, each line of the DoE (or run) has been replicated? Have the different lines of the DoE been performed with the same insert or a new insert has been used for each DoE line? Have the experiments been performed in dry conditions? How length of material has been machined for each conditions or DoE lines (run number)?

The statistical analysis is well developed and the use of an analysis of variance (ANOVA) for such problem is well adequate. As it is difficult to base a model from physical considerations, a quadratic form is finally used.

From the different models obtained by analysis of variance (exit burr, pivot point location, segmentation frequency, maximum nodal temperature, degree of chip segmentation) why the minimization of the exit burr was not carried out under constraint of the other responses ? And then, the FEM and experimental validation of the calculated optimal conditions?

Conclusion

This section clearly points out the results of the research and explains how the research has moved the body of scientific knowledge forward.

Language

The paper is written in a good English but all the document must be checked to eliminate some mistakes.

Author Response

I would like to thank the reviewer for his constructive comments and valuable suggestions. These comments not only encouraged but also guided me to improve the quality of the work. I have followed the instructions carefully and to the best of my understanding. All the additions to the original manuscript are highlighted in red. Rephrasing and rewording are done to improve the logical flow. Hope the revised manuscript will meet the required standard for approval. Please find the point-by-point responses to the reviewers’ comments as:

Comment: English language and style

( ) Extensive editing of English language and style required 
(x) Moderate English changes required 
( ) English language and style are fine/minor spell check required 
( ) I don't feel qualified to judge about the English Language and Style

Response: I have tried my best to refine the level of English to meet the required standard. Complex sentences are simplified and explained for better comprehension in the revised manuscript.

Comment: The title clearly describes the article, but is long enough.

Response: Title is now revised in the revised manuscript.

Comment: Please add a paragraph explaining how the tool edge geometry affects chip segmentation and exit burr since it is the core of the paper. Define the parameters concerning edge preparation which will be studied after (see paper of Denkena et al. DOI: 10.1016/j.procir.2012.04.033).

Response: Related information is now incorporated (L58-L66, L69-L74) in the revised manuscript.

Comment: Table 5, which gives the results of cutting forces and chip segmentation frequencies, needs to be completed giving the results for the 24 DoE lines that have been experimentally performed. These data could be interesting for other authors. In figure 6, add a picture of the experimentally obtained chip as in figure 2. Idem for figure 8. Add a table with the dimensions of the exit burr as table 5 for cutting forces and chip segmentation frequency.

Response:  In Table 5, numerical results of cutting forces and chip segmentation frequencies only for level 3, 4, 15 and 16 are compared with available experimental data results [11]. This comparison is made to validate the numerical model. Whereas, rest of numerical simulations made with various combinations of speed, feed and tool edge geometry (Level 1, 2, 5-14 and 17-24) are merely exploitation of the validated numerical model (with no experimental results found in literature). Same has now been included in the revised manuscript (L210-L217) for more clarity purpose.

Numerical simulation performed with VC = 800 m/min, f = 0.4mm/rev, req = 180 µm (level 23) and results shown in figure 6 (now figure 7 in revised manuscript) and figure 8 (now figure 9 in revised manuscript) is merely exploitation of numerical model (with no experimental results found in literature). While numerical model is validated with findings of the simulation work performed only at level 3, 4, 15 and 16 with available experimental data results [11].

Table 6, is now added in the revised manuscript detailing burr lengths obtained experimentally.

Comment: Mesh density of the order of 20 μm is chosen in “chip separation zone”, chip and upper layer (~ 0.3 mm) of machined workpiece. Why a finer mesh has not been chosen in the chip separation zone (since this zone may be very small of the order of 5 microns)?

Response: Comprehensive detail on selection of mesh density is now included (L126-L133, and new appended Figure-2) in the revised manuscript. While, size of “chip separation zone” vary according to tool edge radius (as per experimental evidence [25], L104-L105). Table 1, provides the details of size of “chip separation zone” chosen for various tool radii.

Comment: The choice of the coefficient of distribution of the friction energy between the chip and the tool (J = 0.5) is questionable and does not take into account the thermal properties of the two materials (aluminium alloy and tool). The use of Coulomb`s friction law is also a simplistic model. Why not a more realistic model (see DOI 10.1007/s00170-016-9149-4)

Response: Related information and justification is now made (L171-L175, L177-L192) in the revised manuscript.

Comment: The paragraph concerning the experimental procedures does not accurately explain how the data were collected. There is no sufficient information present to replicate the research. The article does not identify the procedures followed; the equipment and materials which have been used.

Are the uncoated carbide inserts CCGX 12 04 08-AL 93 H10, the ones used in the experimental part? How has been modified the cutting edge? How many times, each line of the DoE (or run) has been replicated? Have the different lines of the DoE been performed with the same insert or a new insert has been used for each DoE line? Have the experiments been performed in dry conditions? How length of material has been machined for each conditions or DoE lines (run number)?

Response:  Numerical results of cutting forces and chip segmentation frequencies only for level 3, 4, 15 and 16 are compared with available experimental data results [11] and no experimental work has been carried out. Whereas, uncoated carbide inserts CCGX 12 04 08-AL 93 H10 was used for the experimentation carried out by Mabrouki et al [11].  DoE (24 runs) have only been performed numerically in the current work, while only available results from [11] were compared with numerical findings to validate the conceived model. However, SEM was used to get more accurate tool profile (of uncoated carbide inserts CCGX 12 04 08-AL 93 H10 which was used in the experimentation [11]) to be incorporated in defining tool geometrical model for numerical simulation. Same details are now included in revised manuscript (L111-L113, L366)

Comment: From the different models obtained by analysis of variance (exit burr, pivot point location, segmentation frequency, maximum nodal temperature, degree of chip segmentation) why the minimization of the exit burr was not carried out under constraint of the other responses? And then, the FEM and experimental validation of the calculated optimal conditions?

Response: ANOVA for minimization of the exit burr are performed and results are presented in Table 8, and related explanation is made (L397-L408 and L383-L386). Validation of ANOVA results with FEM results is now made (L383-L386) in the revised manuscript.  

Round 2

Reviewer 3 Report

The reviewer agrees with the comments and modifications made to the manuscript.

The paper can be accepted for publication.